# A Review of Dietary (Phyto)Nutrients for Glutathione Support

**DOI:** 10.3390/nu11092073

**Published:** 2019-09-03

**Authors:** Deanna M. Minich, Benjamin I. Brown

**Affiliations:** 1Human Nutrition and Functional Medicine Graduate Program, University of Western States, 2900 NE 132nd Ave, Portland, OR 97230, USA; 2BCNH College of Nutrition and Health, 116–118 Finchley Road, London NW3 5HT, UK

**Keywords:** broccoli, cancer prevention, cruciferous vegetables, glutathione, glutathione S-transferase, green tea, nutrigenomics, phytonutrients, plant-based diet, selenium, vitamins

## Abstract

Glutathione is a tripeptide that plays a pivotal role in critical physiological processes resulting in effects relevant to diverse disease pathophysiology such as maintenance of redox balance, reduction of oxidative stress, enhancement of metabolic detoxification, and regulation of immune system function. The diverse roles of glutathione in physiology are relevant to a considerable body of evidence suggesting that glutathione status may be an important biomarker and treatment target in various chronic, age-related diseases. Yet, proper personalized balance in the individual is key as well as a better understanding of antioxidants and redox balance. Optimizing glutathione levels has been proposed as a strategy for health promotion and disease prevention, although clear, causal relationships between glutathione status and disease risk or treatment remain to be clarified. Nonetheless, human clinical research suggests that nutritional interventions, including amino acids, vitamins, minerals, phytochemicals, and foods can have important effects on circulating glutathione which may translate to clinical benefit. Importantly, genetic variation is a modifier of glutathione status and influences response to nutritional factors that impact glutathione levels. This narrative review explores clinical evidence for nutritional strategies that could be used to improve glutathione status.

## 1. Introduction

Glutathione is a tripeptide (cysteine, glycine, and glutamic acid) found in relatively high concentrations in many bodily tissues [1]. It plays a pivotal role in reducing oxidative stress, maintaining redox balance, enhancing metabolic detoxification, and regulating the immune system [1]. Various chronic, age-related diseases such as those related to neurodegeneration, mitochondrial dysfunction, and even cancer, have been related to suboptimal or deficient glutathione levels [1,2,3]. There is increasing awareness of its utility in mitigating body toxin load through its ability to enhance hepatic conversion and excretion of compounds such as mercury and persistent organic pollutants (POPs) [1,4,5].

As a result, it is possible that supporting the body’s endogenous levels of glutathione would be important for maintaining health and mitigating disease, although clear causal relationships between low glutathione and disease risk remain to be determined. One confounding factor is the complexity of antioxidants, referred to by Halliwell [6] as the “antioxidant paradox”, or the situation in which antioxidants such as glutathione can possess prooxidant activity causing a hormetic effect enabling the body to bolster its endogenous antioxidant defenses. Indeed, redox balance can be the cause or consequence of a disease, and in some cases, it is difficult to know the level at which an antioxidant becomes a prooxidant. Therefore, there is much to understand about the role of glutathione levels in health. 

A factor influencing glutathione status is the degree of variability in an individual’s capacity to produce glutathione, mainly due to genetic variability in enzymes involved in its production and/or regeneration. The enzymes that have received increased attention in the scientific literature and within clinical medicine include glutathione-S-transferase and gamma-glutamyl transferase. Some of these enzymes require nutrient cofactors [1] (Figure 1). With upregulated oxidative stress, malnutrition or increased toxic burden due to exposure to environmental contaminants, there can be even greater need for glutathione [7,8,9]. A list of some proposed disease states related to inadequate glutathione status are listed in Table 1. 

While there may be a need to repair low levels of glutathione, proper balance, rather than excess, is required. For example, glutathione may need monitoring in patients undergoing chemotherapy due to the potential to support chemoresistance [17]. There are challenges in the use of glutathione as a diagnostic biomarker and therapeutic target. Red blood cell (RBC) glutathione is readily available as a clinical assessment but has been found to have wide intra-individual variation [31]. However, this intra-individual variation is relatively stable over time which is a phenomenon likely due to variation in genes regulating glutathione levels in RBCs [32]. Well established reference ranges for RBC glutathione are needed and dynamic, personalized tracking of RBC glutathione may also need to be considered. Another biomarker that may be clinically useful is serum γ-glutamyltransferase (GGT) which is primarily involved in the metabolism of extracellular reduced glutathione and when elevated (above 9 U/L in women and above 14–20 U/L in men) but still within “normal” reference ranges may indicate glutathione depletion and be associated with chronic disease risk [33]. 

## 2. The Role of Gene Deletions and Single Nucleotide Polymorphisms (SNPs) 

There are some common single nucleotide polymorphisms (SNPs) that impact glutathione and associated processes and may subsequently influence disease risk. These code for the enzyme glutathione S-transferase (GST), which conjugate the reduced glutathione to substrates during the detoxification process [34]. *GSTP1* and *GSTM1* do have multiple SNPs (for example, *GSTP1* rs1695, A105G results in an amino acid substitution in codon 105 from valine to isoleucine associated with increased cancer risk), however, the null alleles described and focused on in this review result from gene deletion between the H3 and H5 regions flanking the gene [35]. During times of oxidative stress, GST genes are upregulated. One of the most common polymorphisms, affecting 20% to 50% of certain populations, is an absence of the *GSTM1* gene (*GSTM1*-null), which decreases detoxification ability among other possible outcomes [36]. *GSTT1* (null) and *GSTP1* (AB/BB) are additional polymorphisms related to a reduction in GST activity [37]. Having one or more of these polymorphisms is associated with an increased risk of certain diseases [37], especially when impacted by environmental triggers such as pollution, smoking, heavy metals, and other toxins. 

## 3. Optimizing Glutathione Production with Nutrients

Whether due to the presence of SNPs, gene deletions or heightened physiological need due to exogenous reasons like toxic load, to some extent, glutathione levels may be supported by dietary and/or supplemental nutrients. This review article will attempt to review the salient human clinical literature to support the use of specific foods and nutrients that may increase or maintain optimal glutathione levels.

## 4. Preformed Glutathione

It would seem to be most efficient to administer oral glutathione as a preformed compound to override the effects of potentially inefficient SNPs and related enzymes. However, there has been some debate regarding whether glutathione given orally would be degraded by digestive peptidases [37,38]. In further support of this theory, some studies [39,40,41] have shown no change in glutathione levels or in parameters of oxidative stress despite acute [40,41] or chronic (four weeks) [39] oral glutathione supplementation. 

There is also some evidence to the contrary. One six-month, randomized, double-blinded, placebo-controlled trial [42] found that taking oral glutathione at either 250 or 1000 mg/day led to significant increases in the body stores of glutathione in 54 non-smoking adults in a dose-dependent manner. There was also a decrease in the markers for oxidative stress at six months as indicated through an improvement in the oxidized (GSSG) to reduced (GSH) glutathione ratio in whole blood, in conjunction with favorable increases in natural killer cell cytotoxicity.

While the data on providing oral glutathione are mixed and inconclusive, recent research suggests that when glutathione is administered in liposomal or sublingual forms it may be made more bioavailable and favorably impact systemic glutathione levels.

In a small study [43] with twelve healthy middle-aged non-smoker subjects taking either 500 or 1000 mg per day liposomal glutathione for four weeks, there was a trend towards increasing glutathione levels in a variety of body compartments, although it only became significant in plasma with the 500 mg dose after two weeks, which was also the largest increase at 25%. There was also an improvement in the ratio of oxidized to reduced glutathione, with the largest decrease in this ratio among those taking the higher dose in the first and second weeks. There was also a decrease in the biomarkers for oxidative stress and an improvement in immune markers such as lymphocyte proliferation and natural killer cell activity. 

In another small study [44] with 16 healthy men with cardiovascular risk factors, those with abnormal reactive hyperemia index, which measures peripheral endothelial function and stiffness, experienced a significant reduction in arterial stiffness after taking 100 mg twice daily of a sublingual glutathione. 

Limited data using intravenous glutathione has been documented in patients with Parkinson’s disease. Sechi et al. [45] administered glutathione intravenously (600 mg twice daily for 30 days) to nine individuals with Parkinson’s disease, and reported significant improvements, which lasted for 2–4 months even after ceasing the therapy. A published case report [46] in a 61 year old man with Parkinson’s disease utilized a multi-faceted protocol consisting of a gluten-free diet, along with medications, certain dietary supplements (such as N-acetyl-cysteine and *Silybum*), and glutathione injections (1400 mg) administered twice or three times weekly also reported symptom improvement. 

Finally, preliminary cellular research using glutathione monoesters has been shown to be an effective delivery agent of glutathione due to improved bioavailability [47].

There continues to be debate as to the best delivery system, whether oral, sublingual, liposomal, or intravenous. Intravenous, sublingual, and liposomal delivery can bypass the breakdown that may occur in digestion, and thus may be superior to oral supplementation.

## 5. N-Acetylcysteine (NAC)

Three conditionally essential amino acids, glycine, cysteine, and glutamic acid combine to form glutathione in a two-step biochemical reaction. First, cysteine is conjoined with glutamate through the action of glutamate cysteine ligase to produce gamma-glutamylcysteine, which proceeds to link with glycine via glutathione synthase [48]. Therefore, the human body requires all three amino acids and adequate enzymatic function to make sufficient quantities of glutathione [49,50]. Cysteine is a sulfur amino acid, which might imply that consuming sulfur-rich foods, especially those containing the sulfur amino acids, may also support glutathione synthesis [51,52].

Cysteine is frequently identified as rate-limiting, which provides the rationale of why N-acetylcysteine (NAC) is frequently studied and suggested as a supplement for glutathione support [50], yet a review of the data indicates its use may be inconclusive or equivocal. A systematic review [53] that included twelve clinical trials utilizing NAC supplementation with a specific focus on cognitive markers indicated that there may be some benefit to using NAC in certain populations; however, the studies were too variable in design and outcome to make any definitive conclusion. As a suggestion in future studies, including a genotype segmentation for participants for glutathione-related enzymes such as GST may lead to different findings and assist in investigating who is more primed for an effect. 

In one study [30] on five people with mild to moderate Parkinson’s disease and three controls, a high dose of NAC (3000 mg taken orally twice daily) for a period of four weeks led to an increase of cysteine levels and antioxidant measures, with no commensurate improvement in oxidative stress measurements (4-hydroxynonenal and malondialdehyde) nor did it increase the level of glutathione in the brain. Additionally, some of the participants experienced an exacerbation of Parkinson’s symptoms that were alleviated upon stopping the NAC supplementation.

Another study [54] looking at those with neurodegenerative disorders found that a single intravenous dose of NAC led to an increase of the blood GSH/GSSG ratio and levels of glutathione in the brain. Those who had the greatest percent change in that ratio also had a greater percent change in their levels of glutathione in their brain. While the study was too small and of too short duration to make any conclusions about the role of NAC in these conditions, what is notable is that through intravenous administration of NAC, brain levels could be altered.

A 12-week clinical trial [55] with children (*n* = 31) with autism administered 60 mg/kg/day in three doses (maximum dose of 4200 mg/day) found that although there was no significant impact on the social impairment associated with autism, there was a significant impact on boosting the glutathione levels in the children. Similar to other studies, more needs to be explored as to how NAC influences glutathione levels, factors that modify individual response to supplementation and how symptomatology interrelates. GST polymorphisms may play a role in the efficacy of NAC. In one study [56] investigating the impact of NAC on noise-induced hearing loss in men (*n* = 53) taking 1200 mg per day for 14 days led to a significant reduction of noise-induced temporary threshold shift, or the amount of hearing loss after a period of heightened noise exposure in a work setting relative to baseline, or pre-shift levels. When the participants were grouped according to their GST genotypes, the researchers found that only those with the null genotypes experienced a significant effect from taking NAC. 

Although NAC is promising as a supplement to both boost glutathione levels and potentially mitigate some of the issues related to oxidative stress [57,58], the research is not conclusive [59,60,61,62], and some of the findings are disease specific. There have also been studies with no significant impact by taking NAC. Moreover, it has been suggested that NAC may work synergistically with other supplemental nutrients. For example, it has been postulated that glycine may be as important as cysteine when it comes to glutathione production, especially when concurrently supplemented with NAC [48]. While further studies are required, it may be the better approach to supplement with both cysteine and glycine to see a boost in glutathione, especially among those who may not have adequate quantities of the amino acids or require higher levels of glutathione. 

For example, in a small study [63] with eight healthy elderly adults and a control group of eight younger subjects, after measuring baseline glutathione synthesis in both groups, the older subjects were orally administered 0.81 mmol NAC per kg per day (around 132 mg/kg/day) and 1.33 mmol glycine/kg/day (roughly 100 mg/kg/day) for 14 days. Initially, the older subjects had 55.2% less glycine and 24.4% less cysteine in their red blood cells. They also had a 46.2% lower glutathione level than the controls. However, after supplementation, the glycine levels increased by 117.6% and the cysteine by 55.1%. Furthermore, they had a 94.6% higher glutathione concentration in their red blood cells, which also led to no statistical difference between the young controls and the elderly subjects in their glutathione levels. In addition, they experienced lower plasma oxidative stress and F_2_-isoprostanes. 

The researchers surmised that the typical reduction of glutathione in the elderly was due to a lower supply of glycine and cysteine, the precursors to glutathione synthesis, and that upon supplementation, they had the ability to stimulate synthesis and restore levels. Although impressive, it is important to recognize that this was a small study. It is worthwhile to note that there was also no translation to clinical benefit, as these subjects were healthy.

Finally, it is important to note that NAC has antioxidant properties in addition to being able to provide cysteine for glutathione synthesis. It is unclear if the effects of NAC on oxidative stress are due to these antioxidant properties or due to increased glutathione synthesis.

## 6. Dietary Protein Considerations

Theoretically, impaired protein digestion may also be a limiting factor in ensuring healthy glutathione levels. A lack of or reduced hydrochloric acid production in the gastric mucosa and/or pancreatic enzyme insufficiency would be important to assess in a patient with low plasma albumin and low glutathione levels and/or symptoms of impaired glutathione activity (e.g., fatigue). Hypochlorhydria may, in fact, be more common in the aging population as the gut physiology changes [64], and the use of certain medications can also impact hydrochloric acid levels [65]. Further, oxidative stress (such as seen with low physiological glutathione levels) [66,67] and certain nutrient deficiencies [68] may also contribute to low stomach acid levels.

Since the precursors and foundation of glutathione are amino acids, intake of dietary protein may influence the amino acid pool from which to draw to synthesize glutathione. Changes in protein consumption [69], including reducing protein levels but remaining within safe levels, may alter plasma glutathione synthesis levels contributing to a reduction in antioxidant capacity. In this study, the researchers found that while individuals were able to recover from a reduction in protein (that remained above the lowest amount considered safe) in terms of nitrogen balance, it took longer for the functional changes in glutathione levels to equilibrate. Urinary excretion of 5-L-oxoproline was suggested as a marker to track glutathione kinetics, particularly the availability of glycine.

Although it is not necessary for most people to supplement with protein to meet their daily requirement, one potential beneficial source when additional protein is necessary is whey protein, likely due to its higher cysteine content [70]. In a small study (*n* = 18) of healthy individuals, whey protein supplementation at a dose of 15, 30, or 45 g/day for 14-days resulted in a dose dependent increase in lymphocyte glutathione levels with the 45 g/day dose increasing lymphocyte glutathione by 24% [71]. In another small randomized control study on cancer patients (*n* = 23) [72], consuming 40 g of whey protein isolates in addition to zinc and selenium increased the glutathione levels (11.7%) as well as functional immune markers, including an increase of 4.8% in their immunoglobulin G levels compared with the control group (*n* = 19). Additionally, a small study of patients with Parkinson’s disease [70] found that supplementing with whey protein compared with soy protein led to a significant increase in the glutathione levels in the blood and the GSH/GSSG ratio, although there was no significant impact on the clinical markers of the disease.

There are potentially other amino acids beyond the glutathione precursors that support glutathione synthesis. A few animal studies [73,74] would suggest that serine, a nonessential amino acid, may be helpful in positively influencing glutathione production, potentially through increased cysteine availability and a decrease of hypermethylation. Alternately, serine may support glutathione levels through its metabolism into glycine, one of the precursor amino acids used for glutathione synthesis. 

## 7. Omega-3 Fatty Acids

Chronic inflammation can contribute to oxidative stress and deplete glutathione supply [75]. Due to their involvement in the production of inflammatory and anti-inflammatory prostaglandins, omega-3 fatty acids have been studied for their effects on glutathione levels. In one study [76] taking 4000 mg of omega-3 supplements daily for 12 weeks led to a better GSH–creatine ratio and reduced depressive symptoms in older adults who had a higher risk of developing depression compared with the control group taking a placebo. Another study in patients with Parkinson’s disease found that taking 1000 mg omega-3 fatty acids from flaxseed oil in conjunction with 400 IU of vitamin E for 12 weeks led to an increase in glutathione concentrations as well as total antioxidant capacity and a reduction in the inflammatory marker, high-sensitivity C-reactive protein, and markers of insulin metabolism [77].

One study [78] investigated the impact of GST polymorphisms on the relationship between omega-3 fatty acids and breast cancer risk in post-menopausal Chinese women in Singapore. An increased protective effect of dietary intake of marine-based omega-3 fatty acids was identified in women with genetic polymorphisms for reduced GST activity. The high consumers of marine sources of omega-3 fatty acids with reduced GST activity polymorphisms (*GSTT1*-null genotype) had at least a 64% reduction of risk compared to the low consumer counterparts, with some polymorphisms experiencing an even greater protection.

Finally, in a recent systematic review [79] of nine studies investigating the co-supplementation of omega-3 fatty acids with vitamin E for oxidative stress parameters, no significant changes were seen in glutathione concentrations or select enzymes such as superoxide dismutase and catalase.

Salmon, as a whole food source of omega-3 fatty acids may favorably influence glutathione status. In pregnant women, consumption of two meals of salmon per week from week 20 of gestation increased glutathione concentration [80]. Although a comparison between fish oil capsules and salmon found no significant difference on glutathione status [81].

## 8. Vitamins 

### 8.1. B Vitamins

Riboflavin is a necessary coenzyme for the activity of glutathione reductase, which converts the oxidized glutathione into its reduced form, the compound required for antioxidant function [82]. While there is a paucity of studies to confirm that riboflavin deficiency negatively impacts glutathione levels, there is indication that homocysteine production and methylation processes require riboflavin [83,84]. Since the methylation cycle is closely linked to that of the trans-sulfuration pathways and glutathione metabolism, riboflavin levels could be important. Thus, it is likely that a riboflavin deficiency would impact glutathione function and may even impact the levels in the body. From a biochemical perspective, pantothenic acid (vitamin B5) may also help support glutathione synthesis through its role in ATP production [85]. B12 deficiency [86] is associated with lower glutathione levels.

### 8.2. Vitamin C 

In 48 individuals with ascorbate deficiency [87], taking 500 or 1000 mg per day of vitamin C for 13 weeks led to an 18% increase in lymphocyte glutathione levels compared with placebo. Similarly, in healthy adults following a self-selected vitamin C-restricted diet [88] and an initial week of placebo supplementation, taking 500 mg L-ascorbate per day for weeks two and three and 2000 mg per day for weeks four and five in a six-week trial led to an increased level of glutathione in red blood cells. The lower dose of 500 mg daily led to the most pronounced rise in glutathione levels.

### 8.3. Vitamin E 

Vitamin E supplementation has been studied to a limited extent in diabetic populations subject to higher endogenous oxidative stress levels [89,90]. In type 1 diabetic children [91], vitamin E supplementation (DL-alpha-tocopherol, 100 IU oral daily dose) significantly increased glutathione by 9% and lowered lipid peroxidation (malondialdehyde) by 23% and HbA1c concentrations by 16% in erythrocytes. A similar study [92] in 20 children with type 1 diabetes and 20 healthy controls found that 600 mg/day of vitamin E for three months improved oxidative stress markers and glutathione levels in the diabetic children. In adults (*n* = 54) with diabetic neuropathy [93], the group provided with a vitamin E supplement (800 IU/day) for 12 weeks had significant improvements in cardiometabolic parameters and plasma glutathione levels compared to the group given the placebo.

## 9. Other Nutrients

### 9.1. Alpha-Lipoic Acid

Alpha-lipoic acid is a multifunctional compound in its ability to serve as a direct scavenger of free radical species and to also help in the regeneration of endogenous antioxidants such as glutathione. A variety of clinical trials in diverse populations [94,95,96,97,98,99] would suggest that alpha-lipoic acid could be important for restoring antioxidant capacity. Children with oxidative stress due to protein malnutrition where given either 600 mg reduced glutathione twice daily, 50 mg alpha-lipoic acid twice daily, or 100 mg NAC twice daily for 20 days [100]. Glutathione and alpha-lipoic acid improved survival rates in these children, compared with the control group. HIV-infected adults (*n* = 33) assigned to either 300 mg alpha-lipoic acid three times daily or placebo for six months resulted in elevated blood total glutathione and lymphocyte response in the therapeutic group relative to the control group [101].

### 9.2. Selenium

Selenium is a known antioxidant and cofactor of glutathione peroxidase. In a mouse study [102], selenium supplementation increased the expression and activity of certain glutathione-related enzymes. Another study [103] in 336 healthy adults, (161 blacks, 175 whites) found a positive relationship between selenium levels and selenium supplementation. Despite similar selenium supplementation levels, glutathione levels increased to a greater extent in whites than blacks. It is worthwhile to note that excess selenium may contribute to oxidative stress rather than relieve it and this effect may be related to certain genotypes [104].

### 9.3. Phytonutrients

There can be potentially mixed clinical results from supplementation with supraphysiological doses of antioxidant vitamins and minerals in isolation, separate from their phytonutrient counterparts. One disadvantage may involve disturbing the redox state of a cell towards a predominantly prooxidant status [105]. Therefore, it might seem that one of the safer approaches to fortifying the innate defense against oxidative stress and improving glutathione levels may be best implemented through the diversity and pleiotropism of multiple phytonutrients. In support of this theory, fruit and vegetable intake has been shown to reduce oxidative stress [106], even in intervention studies [107,108,109,110,111]. There needs to be sustainable, creative ways for people to get their daily intake of fruits and vegetables as this quota is not being met by the vast majority [112].

Very few studies [113] have examined the effect of fruit and vegetable intake on glutathione and glutathione-related enzyme levels. High GST activity has been associated with cancer prevention [114]. Previous cell and animal studies [115,116,117,118,119,120] have indicated that certain plant food components (e.g., glucosinolate metabolites and dithiothiones in brassica vegetables; diallyl sulfides from the allium family, citrus-derived limonoids and flavonoids) may upregulate enzymatic activity favorably [121].

One clinical study in 94 subjects monitored diet, lifestyle, *GSTM1* and *GSTT1*-null polymorphisms. Rectal GST enzyme activity was also measured in select participants. The findings revealed that fruit, particularly, citrus fruits were positively associated with greater enzymatic activity. Vegetables, especially the brassica variety, was correlated with GST enzyme activity in the *GSTM1*-plus genotype carriers, but not the *GSTM1*-null individuals. Allium vegetables did not influence enzyme activity. 

### 9.4. Brassica Vegetables

There is a plethora of research to suggest the detoxification and cancer preventative qualities of cruciferous vegetable intake [122,123], especially for cancers related to the gastrointestinal tract [124]. Studies [125,126] have shown that administration of cruciferous-derived compounds, such as sulforaphane, may increase glutathione, glutathione-related enzymes, and even endogenous antioxidant enzymes and inflammatory markers, although results are not always consistent [115]. These compounds may be especially important for individuals with GST polymorphisms. 

The research group at the Fred Hutchinson Cancer Research Center in Seattle, Washington, has published several feeding studies in individuals with varying GST genotypes to investigate whether cruciferous vegetables would influence enzyme activity [117,127,128,129,130,131]. There is limited data to suggest that there is a trend for cruciferous vegetables to be efficacious in increasing GST in those who have GST polymorphisms [117,127] although the results are not always consistent [128,130,132]. A randomized cross-over clinical trial [127] included 33 men and 34 women who ate four different diets each for 14 days: one that was vegetable-free (“basal”), the basal diet supplemented with either a single or double dose of crucifers, and single dose cruciferous plus apiaceous vegetables at a level that was standardized to kilogram body weight. Results indicated that cruciferous vegetables intake (either single or double dose) led to increased GSTA1/2, especially in male subjects with the *GSTM1*-null/*GSTT1*-null genotype. 

In a crossover design [133], young healthy smokers (*n* = 27) consumed steamed broccoli (250 g daily) or a control diet for 10 days; broccoli consumption reduced oxidized DNA by 41% and resistance to hydrogen peroxide-induced DNA strand breaks increased by 23%. Yet, a higher protection was conferred in those with the *GSTM1*-null genotype. Furthermore, another study [124] with 82 cigarette smokers, oral 2-phenethyl isothiocyanate (PEITC), a cruciferous-based compound from watercress, was able to lead to increased detoxification of the volatile carcinogens, benzene and acrolein with an even stronger effect for those with the null genotype of both *GSTM1* and *GSTT1.* No effect of PEITC was seen for those with both genes.

### 9.5. Green Tea

Green tea consumption is associated with reduced rates of certain cancers such as leukemia [134]. In a multicenter case–control study [134] in China with 442 confirmed adult leukemia cases and 442 controls, green tea intake and GST genotypes were assessed. Researchers found not only an inverse association between drinking green tea and adult leukemia risk compared with those who did not drink tea, but that cancer risk reduction was more pronounced in those with the *GSTT1*-null genotype than the *GSTT1*-present carriers. 

Thirty-five obese individuals with metabolic syndrome were randomly assigned to receive one of these interventions for eight weeks: green tea at four cups daily, four cups water daily, or green tea extract (two capsules + four cups water daily) [135]. Blood samples and diet records were collected at baseline and at completion of the study. Both the green tea and green tea extract significantly increased plasma antioxidant capacity and whole blood glutathione compared with the group that only received water. 

### 9.6. Juice Studies

For those for whom eating fruits and vegetables is challenging, drinking juice derived from these foods may provide another healthful option, although some health professionals might be concerned with their simple sugar content. Generally, clinical studies would suggest that drinking fruit and/or vegetable juices confer health benefits, such as improving antioxidant status [136,137,138,139].

Due to the content of polyphenols and the role of these constituents in reducing oxidative stress, some researchers have examined polyphenol-rich juices such as pomegranate and grape juice. In nine elite weightlifters subject to an intensive weightlifting session, pomegranate juice reduced oxidative stress and improved antioxidant enzymatic activity (including a +6.8% increase for glutathione peroxidase) compared with a non-polyphenol placebo [140]. Moreover, in a study in smokers with GST polymorphisms [141] purple grape juice supplementation (480 milliliters daily for eight weeks) had a significant impact, leading to a significant reduction in DNA damage for all participants. Plasma vitamin E was increased in the *GSTM1*-null group, while glutathione levels were increased in the *GSTM1*-present group, and blood vitamin C levels were increased in the *GSTT1*-present genotype, demonstrating the antioxidant impact of the grape juice differed based on the genotype of the participants. The impact on blood pressure also differed based on genotype. Another intervention study [142] in 24 individuals with a crossover design assigned participants to a daily dose of 400 mL conventional grape juice, organic grape juice, or water. While genotypes were not investigated, blood markers for antioxidant enzymes were measured at baseline and then acutely for up to three hours. Drinking both grape juices led to significantly increased glutathione, total antioxidant capacity, catalase, superoxide dismutase, and glutathione peroxidase compared with water. The increase in glutathione peaked at 1 h after consumption. 

Although not related to polyphenols, another study indicated that consuming 300 mL of kale juice daily for six weeks had varying effects based on GST subtype [143], especially in the vitamin C levels and DNA damage, with the *GSTM1*-null genotype experiencing the most favorable impact from the kale juice.

### 9.7. Herbs and Roots

While there is a lack in human clinical trial data, there are several animal studies which would indicate that certain herbs and roots, such as rosemary [144,145,146], turmeric/curcumin [147], milk thistle [148], and *Gingko biloba* [149], may influence glutathione levels. Rosemary extract in the diet of female rats at concentrations of 0.25% to 1.0% by weight resulted in a 3.5- to 4.5-fold increase in hepatic GST. An increase was seen when injected intraperitoneally but to a lesser extent [145]. In an animal study, a turmeric extract and curcumin were shown to increase hepatic glutathione content [150].

### 9.8. Plant Foods that Contain Glutathione

While the focus of this review has been on foods and dietary-derived nutrients for the purpose of supporting antioxidant defenses, primarily by increasing glutathione levels and enzymes related to glutathione’s activity, it is worthwhile to note that there are several foods that contain the thiol-rich compounds, glutathione, NAC, and cysteine (Table 2). Eating a glutathione-supported diet could involve the inclusion of these foods daily, especially the green foods, asparagus, avocado, cucumber, green beans, and spinach. Some preparation tips are listed in Table 3.

While multi-component dietary interventions specifically designed to enhance glutathione status have not yet been studied, it is interesting note that feeding mice a western-style diet impaired hepatic glutathione synthesis and lowered plasma levels [156], and that, in humans, higher adherence to a traditional-Mediterranean style diet is associated with higher plasma glutathione [157]. Food-based nutritional interventions specifically designed to optimize glutathione levels could be a fruitful area for further research. 

## 10. Conclusions

Glutathione is a biomarker associated with disease risk and health status that has potential to translate to an important target relevant to health optimization, disease prevention, and treatment. Although well-established reference ranges for RBC glutathione or GGT are lacking and likely confounded by intra-individual variation, it may be possible to monitor glutathione status using these biomarkers dynamically in a personalized way. Further challenges to clinical translation include clear, causative associations between glutathione and disease risk and clinical outcomes as well as the influence of genetic variation on glutathione status and response to nutritional factors that influence glutathione status. There is also some ambiguity in the evidence exploring the ability of nutritional interventions to enhance glutathione status, with more research needed to clarify optimal dose and delivery forms as well as identification of sub-groups of individuals most likely to respond to particular nutrients or foods.

However, despite the aforementioned challenges it is apparent that optimizing dietary intake of glutathione precursors, co-factors, and whole foods that have been shown to enhance glutathione status or are a source of glutathione would be a relatively simple, low cost, and safe approach that could improve health by optimizing glutathione status in an individual. In a clinical setting this could be implemented with advice to consume foods that have some evidence to suggest they improve glutathione status such as lean protein sources, brassica vegetables, polyphenol-rich fruits and vegetables, herbs and spices, green tea, and omega-3 fatty acid rich-foods such as fish. Dietary supplements may also be useful in certain settings (Table 4). The ability of foods rich in glutathione to influence glutathione status is also an interesting possibility that requires clarification. Multi-component dietary interventions specifically designed to enhance glutathione status represent an exciting opportunity for clinical medicine and future research.

## Figures and Tables

**Figure 1 nutrients-11-02073-f001:**
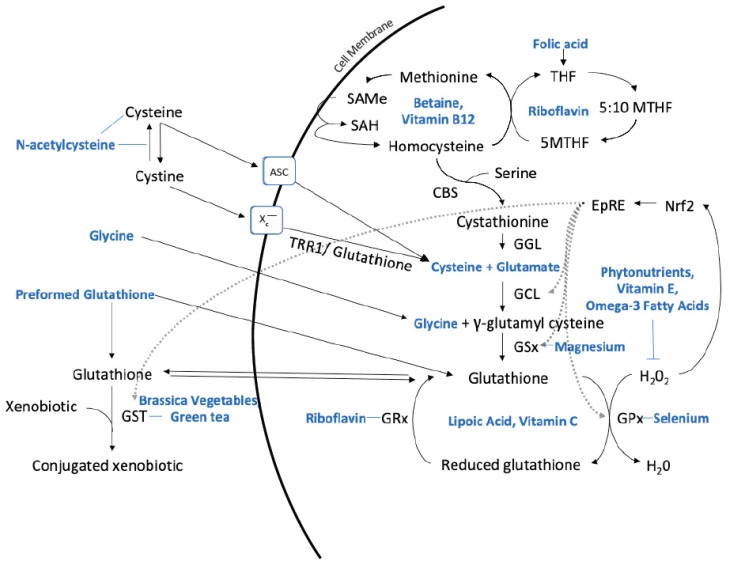
Hepatic synthesis of glutathione and nutritional substrates, co-factors, and other nutrients that influence metabolism. Key: 5-Methyl-tetrahydrofolate (5MTHF), system alanine–serine–cysteine (ASC), cystathionine-β-synthase (CBS), cystathionine gamma-lyase (CGL), electrophile response element (EpRE), glutathione-S-transferase (GST), glutamate cysteine ligase (GCL), glutathione reductase (GRx), glutathione peroxidase (GPx), glutathione synthetase (GSx), hydrogen peroxide (H_2_O_2_), Nuclear factor erythroid factor-2-related factor 2 (Nrf2), S-adenosylmethionine (SAMe), S-adenosylhomocysteine (SAH), tetrahydrofolate (THF), thioredoxin reductase 1 (TRR1), water (H_2_O), cystine/glutamate antiporter system (x_c_^−^). Description: Folic acid is reduced to THF and converted to 5MTHF which can subsequently be transferred to homocysteine and generate methionine. Methionine forms SAMe, which produces SAH from methylation reactions. SAH is hydrolyzed to homocysteine. Homocysteine can either regenerate methionine or be directed to the trans-sulfuration pathway forming cystathionine via the catalytic activity of CBS and serine. CGL cleaves the sulfur–gamma carbon bond of cystathionine, resulting in the release of cysteine which can be used by GCL and GSx to form glutathione. Extracellular cysteine can be either taken up by the cysteine transporter ASC or oxidized to cystine and taken up by system x_c_^−^. N-acetylcysteine can donate cysteine or reduce plasma cystine to cysteine. Intracellular cystine is reduced to cysteine via TRR1 or glutathione. The synthesis of γ-glutamyl cysteine is catalyzed by GCL from cysteine and glutamate, and the addition of glycine to γ-glutamyl cysteine via GSx generates glutathione. GPx catalyzes the reduction of H_2_O_2_ by glutathione and forms reduced glutathione which is then recycled to glutathione by GRx. Glutathione can also form adducts and conjugate xenobiotics via GST. Oxidative stress activates the Nrf2 pathway which induces EpRE-dependent gene expression of enzymes involved in glutathione metabolism, including GCL, GSx, GPx, and GST, to re-establish cellular redox homeostasis. Modified and developed from [10,11,12,13,14].

**Table 1 nutrients-11-02073-t001:** Clinical conditions and diseases associated with glutathione.

Research has found that many chronic diseases are associated with a reduction in glutathione levels, leading to the hypothesis that increasing glutathione levels can help prevent and/or mitigate the progression of these diseases. Below is a list of some of the diseases [2] and issues associated with glutathione dysregulation or deficiency [3]:
• aging [15] and related disorders [3]
• Alzheimer’s disease [16]
• cancer [17]
• chronic liver disease [18]
• cognitive impairment [19]
• cystic fibrosis [20]
• diabetes [21], especially uncontrolled diabetes [22]
• human immunodeficiency virus (HIV)/ acquired immune deficiency syndrome (AIDS) [23]
• hypertension [24]
• infertility in both men and women [25]
• lupus [26]
• mental health disorders [27]
• multiple sclerosis [28]
• neurodegenerative disorders [29]
• Parkinson’s disease [30]

**Table 2 nutrients-11-02073-t002:** Sulfur-rich fruits and vegetables (modified from [151], values reported as mean ± SD (*n* = 3); ND = not detectable). Numbers represent nM/g wet weight (mean ± SD of three samples).

Food	Glutathione	NAC	Cysteine
Asparagus	349 ± 26	46 ± 1	122 ± 1
Avocado	339 ± 10	ND	4 ± 1
Banana	ND	ND	7 ± 0
Broccoli	4 ± 1	ND	ND
Carrot	4 ± 0	ND	ND
Cauliflower	6 ± 1	ND	7 ± 1
Cucumber	123 ± 38	6 ± 1	11 ± 3
Grapefruit	13 ± 3	4 ± 0	15 ± 2
Green Beans	230 ± 2	ND	67 ± 11
Green Pepper	8 ± 1	12 ± 2	9 ± 1
Green Squash	47 ± 11	ND	6 ± 1
Lemon	5 ± 0	4 ± 0	6 ± 0
Mango	59 ± 6	ND	10 ± 0
Orange	5 ± 11	ND	41 ± 2
Papaya	136 ± 12	ND	58 ± 5
Parsley	17 ± 9	9 ± 1	8 ± 1
Potato	5 ± 0	ND	ND
Red Pepper	42 ± 2	25 ± 4	349 ± 18
Spinach	313 ± 33	ND	84 ± 2
Strawberry	39 ± 8	5 ± 1	59 ± 5
Tomato	64 ± 10	3 ± 1	55 ± 3
Yellow Squash	39 ± 8	ND	27 ± 6

**Table 3 nutrients-11-02073-t003:** Preparation tips for sulfur-rich vegetables.

Eat preferably raw or mildly steamed to preserve the integrity of sulfur compounds [152] Refrain from freezing cruciferous vegetables, like broccoli [153] Add powdered mustard seeds during the heating process to increase sulforaphane content [154,155]

**Table 4 nutrients-11-02073-t004:** Summary of nutrients and foods for support of glutathione levels.

Nutrient and Foods	Recommended Dosage
Alpha lipoic-acid	300 mg 3× day; 200–600 mg/day [158]
Brassica vegetables	250 g/day
Curcumin	Doses up to 12 g/day safe; 1–2 g/day found to benefit antioxidant capacity; increased bioavailability with piperine [159]
Fruit and vegetable juices	300–400 mL/day
Glutathione (Liposomal)	500–1000 mg/day [43]
Glutathione (Oral)	500–1000 mg/day [41,42]
Glycine	100 mg/kg/day [63]
Green tea	4 cups/day
N-acetylcysteine	600–1200 mg/day in divided doses, but up to 6000 mg/day have been shown effective in studies [30,53,56,160]
Omega-3 fatty acids	4000 mg/day [76]
Salmon	150 g twice a week [80]
Selenium	247 μg/day of selenium enriched yeast; 100–200 ug/day. Anything above 400 ug/day watch for toxicity [103,160]
Vitamin C	500–2000 mg/day [87,88]
Vitamin E	100–400 IU/day [77,91]
Whey Protein	40 g/day [72]

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
