# Peer review of "A Review of Dietary (Phyto)Nutrients for Glutathione Support"

_nutrients, 2019, doi:10.3390/nu11092073_

Round 1
Reviewer 1 Report
Minich ,DM et al focus on the dietary strategies to optimize glutathione status and they report a wide literature on this topic. Even if in the abstract they remarked that the relationship between glutathione status and disease risk or treatment need to be clarified, they have not discussed this important point. In fact, it has been largely reported that the increase of GSH levels in cancer patients can favour cancer survival and support chemoresistance. Therefore, it is important to take into account this data in order to define the right dietary approach in oncologic patients avoiding the occurrence of side effects.
Minor points
Typography mistakes such as casual relationships in the abstract instead of causal relationships
References in the brackets are wrongly written in all the manuscript : for exemple (3-5) instead of (3-5)
English revision is needed
Author Response
Reviewer #1:
Comment:
Minich ,DM et al focus on the dietary strategies to optimize glutathione status and they report a wide literature on this topic. Even if in the abstract they remarked that the relationship between glutathione status and disease risk or treatment need to be clarified, they have not discussed this important point. In fact, it has been largely reported that the increase of GSH levels in cancer patients can favour cancer survival and support chemoresistance. Therefore, it is important to take into account this data in order to define the right dietary approach in oncologic patients avoiding the occurrence of side effects.
Response:
This is an excellent point. The abstract was amended to include the following sentence:
Abstract (200 words): Glutathione is a tripeptide that plays a pivotal role in critical physiological processes resulting in effects relevant to diverse disease pathophysiology such as reduction of oxidative stress, enhancement of metabolic detoxification, and regulation of immune system function. The diverse roles of glutathione in in physiology are relevant to a considerable body of evidence suggesting that glutathione status may be a novel biomarker and treatment target in various chronic, age-related diseases. Yet, proper personalized balance in the individual is key as well as a better understanding of antioxidants and redox balance. Optimizing glutathione levels has been proposed as a strategy for health promotion and disease prevention, although clear casual relationships between glutathione status and disease risk or treatment remain to be clarified. Nonetheless, human clinical research suggests that nutritional interventions, including amino acids, vitamins, minerals, phytochemicals and foods can have important effects on circulating glutathione which may translate to clinical benefit. Importantly, genetic variation is a modifier of glutathione status and influences response to nutritional factors that impact glutathione levels. This narrative review explores clinical evidence for nutritional strategies that could be used to improve glutathione status.
More explanation was included in this paragraph:
“As a result, it is possible that supporting the body’s endogenous levels of glutathione would be important for maintaining health and mitigating disease, although clear causal relationships between low glutathione and disease risk remain to be determined. One confounding factor is the complexity of antioxidants, referred to by Halliwell as [REF] the “antioxidant paradox,” or the situation in which antioxidants such as glutathione can possess prooxidant activity causing a hormetic effect enabling the body to bolster its endogenous antioxidant defenses. Indeed, redox balance can be the cause or consequence of a disease, and in some cases, it is difficult to know the level at which an antioxidant becomes a prooxidant. Therefore, there is much to understand about the role of glutathione levels in health.”
Additionally, another sentence (lines 71-73) was included to suggest that excessive levels of glutathione may be contraindicated in certain conditions.
“While there may be a need to repair low levels of glutathione, proper balance, rather than excess, is required. For example, glutathione may need monitoring in patients undergoing chemotherapy due to the potential to support chemoresistance.”
Comment:
Typography mistakes such as casual relationships in the abstract instead of causal relationships
Response:
Thank you. This mistake has been fixed and manuscript has been edited for further errors.
Comment:
References in the brackets are wrongly written in all the manuscript : for exemple (3-5) instead of (3-5)
Response:
Thank you. The entire manuscript has been reviewed for this incorrect formatting and corrected.
Comment:
English revision is needed
Response:
Thank you. The entire manuscript has been proofread for grammar and other formatting inconsistencies.
Reviewer 2 Report
Here, Minich and Brown review relevant literature about the effects of dietary nutrients on glutathione support. The review would be bolstered by the inclusion of a section dedicated to the synthesis of glutathione including the genes/proteins necessary for each step of the process. These proteins should be added to figure 1 similar to other important proteins like GPX, GST etc. The authors should also introduce and discuss these different players in more detail. Likewise, they should include a thorough review of the polymorphisms associated with these various genes (GCLC/GCLM/GSS) and how they affect global GSH levels and disease association. It would also be helpful to detail the sources and major sites of GSH biosynthesis in the body. Table 1 lacks a number of diseases/conditions (e.g. hemolytic anemia, increased susceptibility to myocardial infarction, 5-oxoprolinuria, etc) normally associated with decreased glutathione due to polymorphisms in the catalytic or modifier subunits of Glutamate cysteine ligase (GCLC or GCLM respectively) or Glutathione synthase (GSS).
The authors conflate single nucleotide polymorphisms (SNPs) with gene deletion. GSTP1 and GSTM1 do have multiple SNPs (For example, GSTP1 rs1695, A105G results in an amino acid substitution in codon 105 from valine to isoleucine associated with increased cancer risk), however, the null alleles described and focused on in the review result from gene deletion between the H3 and H5 regions flanking the gene (PMID:10975610).
It is important to note that N-Acetyl-Cysteine has antioxidant properties in addition to being able to provide cysteine for GSH synthesis. It is unclear if the effects of NAC on oxidative stress are due to these antioxidant properties or due to increased GSH synthesis. This should be acknowledged.
Page 4, line 105 contains a typo, a parenthesis between non-smoker and subjects – non-smoker) subjects.
Page 5 line 167, the authors write “…led to a significant reduction of noise induced temporary threshold shift.” It is not clear what this means.
Page 6 – line 221 the authors indicate serine may positively influence GSH production through increased cysteine availability and decreased hypermethylation. Serine can be metabolized into glycine. Another possibility is that Serine metabolism support GSH synthesis by providing Glycine, the final amino acid in GSH. This should be mentioned as another possibility.
Reference i, ii, v, vi, and ix are the same reference.
Reference xxxix and xl are duplicated.
Author Response
Reviewer #2:
Comment:
Here, Minich and Brown review relevant literature about the effects of dietary nutrients on glutathione support. The review would be bolstered by the inclusion of a section dedicated to the synthesis of glutathione including the genes/proteins necessary for each step of the process. These proteins should be added to figure 1 similar to other important proteins like GPX, GST etc. The authors should also introduce and discuss these different players in more detail. Likewise, they should include a thorough review of the polymorphisms associated with these various genes (GCLC/GCLM/GSS) and how they affect global GSH levels and disease association. It would also be helpful to detail the sources and major sites of GSH biosynthesis in the body. Table 1 lacks a number of diseases/conditions (e.g. hemolytic anemia, increased susceptibility to myocardial infarction, 5-oxoprolinuria, etc) normally associated with decreased glutathione due to polymorphisms in the catalytic or modifier subunits of Glutamate cysteine ligase (GCLC or GCLM respectively) or Glutathione synthase (GSS).
Response:
To address this Figure 1 has been significantly revised to include several more proteins, enzymes and the Nrf2 system.
Table 1 is an example of some common disease associations, and not intended as an in-depth review or exhaustive list.
Comment:
The authors conflate single nucleotide polymorphisms (SNPs) with gene deletion. GSTP1 and GSTM1 do have multiple SNPs (For example, GSTP1 rs1695, A105G results in an amino acid substitution in codon 105 from valine to isoleucine associated with increased cancer risk), however, the null alleles described and focused on in the review result from gene deletion between the H3 and H5 regions flanking the gene (PMID:10975610).
Response:
Thank you for pointing out this distinction. The section on SNPs had had this change in verbiage to the header:
The Role of Gene Deletions and SNPs
Furthermore, this sentence has been added to the section on genes:
“GSTP1 and GSTM1 do have multiple SNPs (for example, GSTP1 rs1695, A105G results in an amino acid substitution in codon 105 from valine to isoleucine associated with increased cancer risk), however, the null alleles described and focused on in this review result from gene deletion between the H3 and H5 regions flanking the gene.”
Sprenger R, Schlagenhaufer R, Kerb R, Bruhn C, Brockmöller J, Roots I, Brinkmann U. Characterization of the glutathione S-transferase GSTT1 deletion: discrimination of all genotypes by polymerase chain reaction indicates a trimodular genotype-phenotype correlation. Pharmacogenetics. 2000 Aug;10(6):557-65.
Comment:
It is important to note that N-Acetyl-Cysteine has antioxidant properties in addition to being able to provide cysteine for GSH synthesis. It is unclear if the effects of NAC on oxidative stress are due to these antioxidant properties or due to increased GSH synthesis. This should be acknowledged.
Response:
Thank you for this excellent point. This sentence has been added to the section on NAC:
“Finally, it is important to note that NAC has antioxidant properties in addition to being able to provide cysteine for glutathione synthesis. It is unclear if the effects of NAC on oxidative stress are due to these antioxidant properties or due to increased glutathione synthesis.”
Comment:
Page 4, line 105 contains a typo, a parenthesis between non-smoker and subjects – non-smoker) subjects.
Response:
Thank you for pointing out this typo. It has been fixed.
Comment:
Page 5 line 167, the authors write “…led to a significant reduction of noise induced temporary threshold shift.” It is not clear what this means.
Response:
Thank you for requesting further clarification. This sentence has been amended to include a definition of TTS in order to better understand the study outcome:
“In one study (66) investigating the impact of NAC on noise-induced hearing loss in men (n=53) taking 1,200 mg per day for 14 days led to a significant reduction of noise-induced temporary threshold shift, or the amount of hearing loss after a period of heightened noise exposure in a work setting relative to baseline, or pre-shift levels.”
Comment:
Page 6 – line 221 the authors indicate serine may positively influence GSH production through increased cysteine availability and decreased hypermethylation. Serine can be metabolized into glycine. Another possibility is that Serine metabolism support GSH synthesis by providing Glycine, the final amino acid in GSH. This should be mentioned as another possibility.
Response:
Thank you for pointing out this important fact. An additional sentence has been added to that paragraph stating that serine metabolism may support GSH via its metabolism to glycine, one of the precursor amino acids used for GSH synthesis.
“There are potentially other amino acids beyond the glutathione precursors that support glutathione synthesis. A few animal studies (84, 85) would suggest that serine, a nonessential amino acid, may be helpful in positively influencing glutathione production, potentially through increased cysteine availability and a decrease of hypermethylation. Alternately, serine may support glutathione levels through its metabolism into glycine, one of the precursor amino acids used for glutathione synthesis.”
Comment:
Reference i, ii, v, vi, and ix are the same reference.
Reference xxxix and xl are duplicated.
Response:
Thank you for this point. All references have been reviewed and duplicates have been removed.
Reviewer 3 Report
The review informs how nutrition can affect glutathione levels and redox status as well as glutathione transferases in the body. For the most part, the provided information about nutrition is useful. However, the molecular effects are unclear, and incompletely explained. The authors should provide a better and concise description of the major strategies to improve glutathione levels and redox status by nutrition. ()
Specific comments:
The first half of the review mostly focuses on neurological disorders.
Abstract: This reviewer respectfully disagrees that glutathione is a novel biomarker. It has been targeted with acetyl-cysteine in clinics for a quite long time. Also, one of the most important functions, which is often overseen, glutathione is the cellular redox buffer.
The format of the citations is inconsistent.
Figure 1: Please make sure there is uniform formatting. Not all items are capitalized (e.g., Fatty Acids).
Page 2, line 46: glutathione peroxidase is misspelled.
Page 2, line 53: Deregulation or dysregulation?
Also, GTT is not a novel biomarker and used in many clinical studies.
Page 3, line 82. There are many different GSTs, and they can form heterodimers. This paragraph is slightly misleading and superficial. Also, the authors refer several times to effect on GSTs and glutathione, but it is not clear how they affect each other and how this is relevant for the outcome of these studies.
There is some literature by Meister et al. and application of cell-permeable glutathione ethyl ester.
The authors should mention that glutathione is also used as a post-translational protein modification, which participates in intracellular signaling. Thus influencing GSH levels and the cellular redox status may affect signaling and gene expression (e.g., NRF2 system).
Does it appear that on page 5 the font size changed?
Magnesium is a cofactor of many enzymes and the relation to GSH levels and oxidation unclear. This paragraph should be removed.
Page 10, line 387: “genetic type,” Better use genotype.
The liver is one of the major organs regulating plasma cysteine and glutathione levels. How will nutrition affect liver function, and what can be the potential effects on other organs? Also, GSTs are very prominent in the liver for detoxification reactions (Phase II), but also redox signaling. May some of the nutrients such as polyphenols stimulate GSH–synthesis via the NRF2 system?
Author Response
Reviewer #3:
Comment:
The review informs how nutrition can affect glutathione levels and redox status as well as glutathione transferases in the body. For the most part, the provided information about nutrition is useful. However, the molecular effects are unclear, and incompletely explained. The authors should provide a better and concise description of the major strategies to improve glutathione levels and redox status by nutrition. ()
Response:
It is unclear exactly what the reviewer requires. The paper focuses specifically on the ability of nutritional factors to increase glutathione in a clinical setting, detailed discussion of molecular mechanisms of each intervention (i.e. nutrient or food) may be beyond the scope of this review.
Comment:
The first half of the review mostly focuses on neurological disorders.
Response:
Table 1. details a number of non-neurological disorders.
Comment:
Abstract: This reviewer respectfully disagrees that glutathione is a novel biomarker. It has been targeted with acetyl-cysteine in clinics for a quite long time. Also, one of the most important functions, which is often overseen, glutathione is the cellular redox buffer.
Response:
Thank you for this excellent point. This verbiage (“glutathione as a novel biomarker”) has been removed entirely from the manuscript and the verbiage regarding “glutathione as a cellular redox buffer” has been added to the abstract.
Comment:
The format of the citations is inconsistent.
Response:
Thank you for this point. All citations have been reviewed and corrected for formatting.
Comment:
Figure 1: Please make sure there is uniform formatting. Not all items are capitalized (e.g., Fatty Acids).
Response:
The authors appreciate your attention to detail. The items listed in the caption have been consistently capitalized. “Fatty acids” was not capitalized as the adjective preceding it (“Omega”) was capitalized, similar to “Brassica vegetables”.
Comment:
Page 2, line 46: glutathione peroxidase is misspelled.
Response:
Thank you for pointing out this typo. It has been corrected.
Comment:
Page 2, line 53: Deregulation or dysregulation?
Response:
Thank you for this clarification. The text has been changed to “dysregulation”.
Comment:
Also, GTT is not a novel biomarker and used in many clinical studies.
Response:
Thank you. All references to glutathione in any form as being a “novel biomarker” have been removed.
Comment:
Page 3, line 82. There are many different GSTs, and they can form heterodimers. This paragraph is slightly misleading and superficial. Also, the authors refer several times to effect on GSTs and glutathione, but it is not clear how they affect each other and how this is relevant for the outcome of these studies.
Response:
I think this would be best addressed by 1) acknowledging that there are many different GSTs, and they can form heterodimers in the text. 2) countering that the reviews primary aim is to explore the effect of nutrients on glutathione status in a clinical setting, the authors feel that molecular interactions between GSTs and glutathione related to each study are beyond the scope of this review.
Comment:
There is some literature by Meister et al. and application of cell-permeable glutathione ethyl ester.
Response:
Thank you for bringing this point to our attention. A sentence has been included in the section on preformed glutathione and the Meister reference was used as the citation:
“Finally, preliminary cellular research using glutathione monoesters has been shown to be an effective delivery agent of glutathione due to improved bioavailability.”
Comment:
The authors should mention that glutathione is also used as a post-translational protein modification, which participates in intracellular signaling. Thus influencing GSH levels and the cellular redox status may affect signaling and gene expression (e.g., NRF2 system).
Response:
The Nrf2 pathway has been added to the figure, with mention of genes influenced by Nrf2 in the image description.
Comment:
Does it appear that on page 5 the font size changed?
Response:
Thank you. All font sizes have been made to be consistent in the revised manuscript.
Comment:
Magnesium is a cofactor of many enzymes and the relation to GSH levels and oxidation unclear. This paragraph should be removed.
Response:
This paragraph was deleted from the revised manuscript.
Comment:
Page 10, line 387: “genetic type,” Better use genotype.
Response:
This requested change was made in the revised manuscript.
Comment:
The liver is one of the major organs regulating plasma cysteine and glutathione levels. How will nutrition affect liver function, and what can be the potential effects on other organs? Also, GSTs are very prominent in the liver for detoxification reactions (Phase II), but also redox signaling. May some of the nutrients such as polyphenols stimulate GSH–synthesis via the NRF2 system?
Response:
The Nrf2 pathway has been added to the figure, with mention of genes influenced by Nrf2 in the image description. While it is an excellent point that nutritional factors that influence the Nrf2 pathway would indeed include glutathione metabolism, we feel that this topic has been sufficiently reviewed elsewhere and is beyond the scope of this review.
Reviewer 4 Report
This is an interesting review summarising the dietary nutrients that can support glutathione concentration (and/or production). It is a well-written manuscript and provides a brief description of many aspects of GSH production. Please, find next some comments on the paper:
The authors make some arguments throughout the paper about the role of oxidative stress in health and disease and how antioxidant treatments may be of clinical help. In my point of view, this perspective underscores the complex biology of redox processes. I mean that oxidative stress can be either the cause or the consequence of a disease (or even just an epiphenomenon non-related to the disease per se). I think that this issue (see for instance the "antioxidant paradox" of Prof. Halliwell published in 2013; PubMedID:22420826) should be clarified in the revised version.
Personalised nutrition is probably one of the most fast growing concepts in the nutrition field. The authors nicely present the role of SNPs in the respective part of the manuscript. However, malnutrition as a source of the inter-individual differences and for the antioxidant deficiencies observed in the general population or in athletes has been neglected (see for instance PubMedID:30256898).
Tables and the reference list should be redesigned. Especially the reference list in its current form is almost impossible to be followed throughout the paper (use of latin numbers and some papers cited under different citation numbers).
Author Response
Reviewer #4:
Comment:
The authors make some arguments throughout the paper about the role of oxidative stress in health and disease and how antioxidant treatments may be of clinical help. In my point of view, this perspective underscores the complex biology of redox processes. I mean that oxidative stress can be either the cause or the consequence of a disease (or even just an epiphenomenon non-related to the disease per se). I think that this issue (see for instance the "antioxidant paradox" of Prof. Halliwell published in 2013; PubMedID:22420826) should be clarified in the revised version.
Response:
Your point is well-taken. The excellent reference you suggested has been included in this paragraph:
“As a result, it is possible that supporting the body’s endogenous levels of glutathione would be important for maintaining health and mitigating disease, although clear causal relationships between low glutathione and disease risk remain to be determined. One confounding factor is the complexity of antioxidants, referred to by Halliwell as [REF] the “antioxidant paradox,” or the situation in which antioxidants such as glutathione can possess prooxidant activity causing a hormetic effect enabling the body to bolster its endogenous antioxidant defenses. Indeed, redox balance can be the cause or consequence of a disease, and in some cases, it is difficult to know the level at which an antioxidant becomes a prooxidant. Therefore, there is much to understand about the role of glutathione levels in health.”
Comment:
Personalised nutrition is probably one of the most fast growing concepts in the nutrition field. The authors nicely present the role of SNPs in the respective part of the manuscript. However, malnutrition as a source of the inter-individual differences and for the antioxidant deficiencies observed in the general population or in athletes has been neglected (see for instance PubMedID:30256898).
Response:
The authors contend that addressing multiple patient types is beyond the scope of this review.
Comment:
Tables and the reference list should be redesigned. Especially the reference list in its current form is almost impossible to be followed throughout the paper (use of latin numbers and some papers cited under different citation numbers).
Response:
Thank you for pointing out this fact. Tables and references have been reformatted and redesigned to accommodate.
Round 2
Reviewer 1 Report
The authors have adequately answered to reviewer’s suggestions and manuscript has been improved.
Author Response
Thank you.
Reviewer 2 Report
The authors have sufficiently addressed my concerns.
Author Response
Thank you
Reviewer 3 Report
There are now three figures in the text, which is confusing.
Otherwise the manuscript has been improved and most reviewers comments have been addressed.
Is the amino acid transporter needed for intracellular uptake of acetyl-cysteine?
Author Response
Reviewers comment:
[Reviewer 3, round 2].
Is the amino acid transporter needed for intracellular uptake of acetyl-cysteine?
Authors response:
Intracellular cysteine levels are tightly regulated to avoid its cytotoxic action, and the cellular uptake of cysteine is thought to be limited for this reason [1]. Uptake of extracellular cysteine into hepatocytes is largely specific to one transport route, the transport system alanine-serine-cysteine (ASC) [2]. There is an alternative pathway by which cysteine is oxidized to cystine and taken up by system xc− [3]. These pathways are both depicted in the diagram, so no changes are required.
Stipanuk MH, Dominy JE Jr, Lee JI, Coloso RM. Mammalian cysteine metabolism: new insights into regulation of cysteine metabolism. J Nutr. 2006 Jun;136(6 Suppl):1652S-1659S. Kilberg MS, Handlogten ME, Christensen HN. Characteristics of system ASC for transport of neutral amino acids in the isolated rat hepatocyte. J Biol Chem. 1981 Apr 10;256(7):3304-12. Lee J, Kang ES, Kobayashi S, Homma T, Sato H, Seo HG, Fujii J. The viability of primary hepatocytes is maintained under a low cysteine-glutathione redox state with a marked elevation in ophthalmic acid production. Exp Cell Res. 2017 Dec 1;361(1):178-191.The revised graphic is attached.

Reviewer 4 Report
Great effort!
No further comments.
Author Response
Thank you